# Genome-Wide Identification and Analysis of WRKY Gene Family in *Melastoma dodecandrum*

**DOI:** 10.3390/ijms241914904

**Published:** 2023-10-05

**Authors:** Ruonan Tang, Yunjun Zhu, Songmin Yang, Fei Wang, Guizhen Chen, Jinliao Chen, Kai Zhao, Zhongjian Liu, Donghui Peng

**Affiliations:** 1Key Laboratory of National Forestry and Grassland Administration for Orchid Conservation and Utilization, College of Landscape Architecture and Art, Fujian Agriculture and Forestry University, Fuzhou 350002, China; trn7020@163.com (R.T.); wsxx54@sina.com (Y.Z.); songminyang3199@163.com (S.Y.); wangfei_icon@163.com (F.W.); cgz1020@126.com (G.C.); fjchenjl@126.com (J.C.); zhaokai@fjnu.edu.cn (K.Z.); zjliu@fafu.edu.cn (Z.L.); 2College of Life Sciences, Fujian Normal University, Fuzhou 350117, China

**Keywords:** *Melastoma dodecandrum*, WRKY gene family, growth and development

## Abstract

WRKY is one of the largest transcription factor families in plants and plays an important role in plant growth and development as well as in abiotic and biological stresses. However, there is little information about the WRKY family in *Melastoma dodecandrum*. In this study, 126 WRKY members were identified in *M. dodecandrum*. According to phylogenetic analysis, they were divided into three major groups, and group II was further divided into five subgroups. *MedWRKY* genes were unevenly distributed on 12 chromosomes. Additionally, the gene structure and sequence composition were similar within the same group and differed between groups, suggesting their functional diversity. The promoter sequence analysis identified a number of cis-acting elements related to plant growth and development, stress response, and secondary metabolite synthesis in the WRKY gene family. The collinearity analysis showed that gene replication events were the main driving force of *MedWRKY* gene evolution. The transcriptome data and RT-qPCR analysis suggested that *MedWRKY* genes had higher expression in the roots and ripe fruit of *M. dodecandrum*. In short, this paper lays a foundation for further study of the functions and molecular mechanism of *M. dodecandrum* WRKY gene family.

## 1. Introduction

During growth and development, plants are often affected by environmental changes, including biotic and abiotic stresses, such as fungi, bacteria, drought, and cold. Therefore, plants induce the expression of many genes in the face of threats, which may be more beneficial to plant survival, by activating or repressing genes [1]. The WRKY family is one of the largest families of transcriptional regulators in plants. It is mainly involved in plant growth and development and plays an important role in abiotic and biotic stress tolerance. The most remarkable structural feature of WRKY proteins is that they have one~two WRKY structural domains, consisting of approximately 60 amino acid residues, and also possess the highly conserved WRKYGQK structural domain [2,3]. Based on the variation of WRKY structural domains and zinc finger motifs, they can be classified into three major groups (Group I, Group II, and Group III). Group I contains two WRKY structural domains and Group II and Group III both have one WRKY structural domain. Both Group I and Group II contain C2H2 zinc finger motifs, while Group III has the C2HC motif [4].

The first WRKY transcription factor was identified in sweet potato [5] in 1994, and this was followed by studies of WRKY genes from *Oryza sativa* [1], *Arabidopsis thaliana* [6], *Triticum aestivum* [7], *Gossypium* [8], *Manihot esculenta* [9], *Ananas comosus* [10], *Acer truncatum* [11], *Nymphaea colorata* [12] and other plants. WRKY transcription factors play an important role in signaling and expression regulation during biotic and abiotic stress [13]. In banana, *MaWRKY26* activates the biosynthesis of jasmonic acid, enhancing the cold resistance of the fruit [14]. In *Arabidopsis*, overexpression of wheat *TaWRKY93* can enhance plant tolerance to salt, drought, and low-temperature stress [15]. WRKY transcription factors are also involved in the regulation of plant growth and development. For example, in *Arabidopsis*, overexpression of the soybean *WRKY16* gene promotes root growth [16]. *OsWRKY05* accelerates the aging process of rice leaves by regulating the abscisic acid biosynthesis pathway [17]. WRKY transcription factors have also been shown to regulate the production of some secondary metabolites [13]. In rice, *OsWRKY53* positively regulates brassinolide signaling and mediates crosstalk between this hormone and other signaling pathways [18]. In cabbage, *BrWRKY06* inhibits the synthesis of gibberellin in vivo by repressing the expression of *BrKAO2* and *BrGA20ox2* [19].

*M. dodecandrum* is a species of the *Melastoma* in the family Melastomataceae, which is abundant in the wild and mainly distributed in the southern regions of the Yangtze River in China [20]. It is an attractive plant with a long flowering time, lovely blossoms, and strong adaptability, and is also a healthy food with significant medicinal values [21]. At present, research on *M. dodecandrum* is mainly focused on chemical composition, pharmacological activity, propagation and breeding, and garden applications. The WRKY gene family has been extensively studied in many plants. However, studies on the WRKY gene family in *M. dodecandrum* are still limited. Since WRKY transcription factors play an important role in various physiological activities, there is a need to systematically study the WRKY family in *M. dodecandrum*.

## 2. Results

### 2.1. Identification and Characterization of MedWRKY Gene Family 

We identified a total of 126 WRKY members in *M. dodecandrum*, which were named MedWRKY1~MedWRKY126. The results of physicochemical properties are shown in Appendix A. Among the 126 MedWRKY proteins, MedWRKY112 was the smallest protein with 110 amino acids (aa) and the largest was MedWRKY70 with 816 amino acids (aa), corresponding to protein molecular weights of 12.14 k Da and 88.78 k Da, respectively. MedWRKY isoelectric points (PI) ranged from 4.91 (MedWRKY4) to 9.99 (MedWRKY6) with a mean isoelectric point of 7.38, which was alkaline. The aliphatic index ranged from 36.46 (MedWRKY64) to 79.95 (MedWRKY43). The results of the instability index showed that all the remaining proteins were unstable, except for MedWRKY54, MedWRKY82, and MedWRKY96. The grand average of hydropathicity (GRAVY) of all 126 MedWRKY proteins was negative, indicating their strong hydrophilicity. Subcellular localization predictions showed that the majority of MedWRKY proteins were localized in the nucleus, indicating that they function primarily in the nucleus.

### 2.2. Multiple Sequence Alignment, Phylogenetic Analysis of MedWRKY Gene Family

Phylogenetic tree construction of protein sequences of the WRKY family of *M. dodecandrum* and *A. thaliana* was performed by MEGA6.0 software. According to the grouping of AtWRKY, the 126 MedWRKYs could be divided into three groups, including group I (26), II (80), and III (20). Among them, group II can be further divided into five major subgroups, including II-a (11), II-b (18), II-c (28), II-d (10), and II-e (13) (Figure 1). II-a and II-b, and II-d and II-e, were closely distributed on the phylogenetic tree, while the distribution of II-c was closer to that of group I, indicating that the two groups were more closely related.

The phylogenetic relationship of the MedWRKYs was analyzed by multiple sequence alignment of the WRKY structural domains. The results of multiple sequence alignment are shown in Figure 2. A highly conserved WRKY motif was present in the MedWRKYs. However, the WRKYGQK sequences in some MedWRKYs were mutated: for example, MedWRKY4, MedWRKY5, MedWRKY40, and MedWRKY84 were mutated to WRKYGKK. The R in the WRKYGQK heptapeptide structural domain of MedWRKY97 was mutated to M, forming the WMKYGQK heptapeptide structural domain. The R in the WRKYGQK heptapeptide structural domain of MedWRKY82, MedWRKY112, and MedWRKY115 was mutated to K, forming the WKKYGQK heptapeptide structural domain. The MedWRKYs that were mutated were mostly in II-c, indicating that the conserved sequence of II-c was prone to mutation. In addition, there was a loss of the structural domain of MedWRKY32 in II-b. Based on the comparison results, we found that although WRKYGQK heptapeptide sequences were highly conserved among MedWRKYs, most had a low sequence similarity outside the structural domain.

### 2.3. Conserved Motif, Gene Structure Analysis of MedWRKY Gene Family

In order to better understand the similarity and diversity of the protein motif compositions, the online software MEME (accessed on 29 March 2023) was used to analyze the conserved motif of MedWRKYs. Ten different motifs were identified and named Motif 1~Motif 10. Motif 1 and Motif 7 contained conservative WRKYGQK domains (Figure 3).

Among the 126 MedWRKY members, each member contained three to eight motifs. Motif 1, Motif 2, and Motif 4 were present in almost every MedWRKY. Some motifs were present within specific groups: for example, Motif 9 was only present in I, Motif 5 was only present in II-a and II-b, and Motif 10 was specific to II-b. In general, the conserved motifs of these proteins within the same group were similar in composition, indicating that MedWRKYs might have similar functions (Figure 4A).

To further understand the characterization of the *M. dodecandrum* WRKY family, the gene structure of the *MedWRKY* genes was analyzed. As shown in Figure 4B, 126 *MedWRKYs* contained 2 to 13 exons and 0 to 12 introns, and the number of exons and introns varied among genes. Among them, *MedWRKY32* had the highest number of exons and introns with 13 and 12, respectively, while *MedWRKY6* and *MedWRKY36* had no introns. This suggests that the *MedWRKY* genes might have undergone exon and intron loss and gain events during evolution. The analysis indicated that genes with similar structures were generally clustered in the same class.

### 2.4. Promoter Sequences Analysis of MedWRKY Gene Family

Analysis of the promoter sequences of 1500 bp upstream showed that the promoter cis-acting elements can bind different types of transcription factors, including bZIP, SBP, TCP, and so on (Appendix A). In this study, we analyzed and displayed 12 transcription factor binding sites, such as AT-HOOK, bHLH, bZIP, CATA, C2H2, etc. Among them, ZF-HD (2805), bZIP (1718), and bHLH (2030) had the largest number, accounting for about 58% of the total. The lowest number was GATA (201), TBP (246), and Dehydrin (266), accounting for only 6% of the total. All *MedWRKYs* contained bZIP and ZF-HD. With the exception of *MedWRKY49*, the rest of the genes contained BHLH. About 94.5% of the *MedWRKYs* contained Trihelix, AP2, ERF, Dehydrin, and C2H2. Additionally, 67% of *MedWRKYs* contained transcription factor binding sites associated with TBP and GATA (Figure 5).

To further understand the function of the *MedWRKY* genes, cis-acting elements in all *MedWRKYs* promoters were analyzed. The results showed that the promoter region of *MedWRKY* genes contained multiple cis-acting elements (Appendix A). These elements included light-responsive elements (G-box, GT1-motif, and Sp1), MeJA-responsive elements (TGACG motif and CGTCA motif), abscisic acid-responsive elements (ABRE), low-temperature responsive elements (LTR), auxin-responsive elements (TGA-element and AuxRR-core), gibberellin responsive elements (TATC-box), and drought-responsive elements (MBS). Among them, the number of light-responsive elements was the largest (665), followed by MeJA-responsive elements (482) and abscisic acid-responsive elements (385) (Figure 6).

### 2.5. Chromosome Localization and Collinearity Analysis of MedWRKY Gene Family

The 126 *MedWRKYs* were unevenly distributed on 12 chromosomes, and seven other genes (*MedWRKY43*, *44*, *45*, *117*, *118*, *123*, and *126*) were located on consecutive clonal lines and were not shown. Among them, the most genes were distributed on chromosome LG01 with nineteen and the least on LG04 with four (Figure 7). It is known from the figure that intragenomic collinearity analysis contained a large number of intrachromosomal duplication events and interchromosomal duplication events, illustrating that gene duplication events were the main driving force of *MedWRKY* evolution.

To further infer the phylogenetic mechanism of *MedWRKY* genes, comparative co-linear maps were constructed between *M. dodecandrum* and *A. thaliana*, *E. grandis,* and *O. sativa*. The result showed that *M. dodecandrum* and *E. grandis* had the most homologous WRKY genes with 118 pairs, and *A. thaliana* had 94 pairs of homologous WRKY genes. *M. dodecandrum* and *O. sativa* had the fewest homologous WRKY genes, with only 14 pairs (Figure 8).

### 2.6. Expression Level Analysis of MedWRKY Gene Family

The expression level of the WRKY gene family in different tissues, such as root, stem, leaf, mature flower, and ripe fruit, were analyzed. As shown in Appendix A, each *MedWRKY* gene was expressed in at least one tissue, except *MedWRKY10* and *MedWRKY42*. *MedWRKYs* had different levels of expression in different tissues (Figure 9). *MedWRKY41*, *MedWRKY53*, *MedWRKY60*, *MedWRK61*, *MedWRK119,* and *MedWRKY125* were highly expressed in root tissue and showed low expression in other tissues. *MedWRKY14*, *MedWRKY35*, *MedWRKY65,* and *MedWRKY114* were significantly expressed in ripe fruit tissue and exhibited low expression in other tissues. *MedWRKY32* was moderately expressed in stem, leaf, and ripe fruit tissue, and showed low expression in mature flower tissue. High expression of *MedWRKY59* was observed in mature flower and ripe fruit tissue, but a low expression was observed in other tissues. *MedWRKY91* showed high expression in root, leaf, and mature flower tissue, but low expression was noticed in stem and ripe fruit tissue. Compared to other tissues, the *MedWRKY* genes had higher expression levels in the root and ripe fruit tissue, and the expression levels of the *MedWRKY* genes in the stem tissue were generally lower.

### 2.7. RT-qPCR Analysis of MedWRKY Gene Family

To verify the accuracy of transcriptome data, 11 genes were selected for RT-qPCR analysis (Appendix A). The results showed that the expression of 11 *MedWRKY* genes was high in the root tissue and low in the stem tissue, and the expression trend of these genes was basically consistent with the transcriptome data, indicating that the transcriptome data analysis results were reliable (Figure 10). According to transcriptome data, *MedWRKY53* showed the highest expression in root tissue and *MedWRKY65* showed the highest expression in ripe fruit tissue. However, *MedWRKY53* and *MedWRKY65* exhibited the highest expression in leaf tissue based on RT-qPCR results. These differences might be caused by imperfect correlation between RNA sequencing and RT-qPCR samples.

## 3. Discussion

WRKY genes are a large family of transcription factors that are ubiquitous in all plants and play an essential regulatory role in plant growth [22]. However, information about the WRKY gene family in *M. dodecandrum* is not known, and the publication of *M. dodecandrum* genomic data provides a valuable tool for genome-wide analysis of the WRKY gene family.

In this study, a total of 126 WRKY genes were identified in *M. dodecandrum,* which were unevenly distributed on 12 chromosomes. The number of *MedWRKY* genes was much higher than that of chickpea [23] (70), *Arabidopsis* [6] (75), rice [1] (102), and maize [24] (119), which may be due to the large-scale replication event of *MedWRKYs* during the evolutionary process. Previous studies have shown that *M. dodecandrum* experienced four WGD events: one γ-event shared with most eudicots, one event shared with Myrtales, and two events specific to *M. dodecandrum* [25]. Physicochemical characterization showed that MedWRKY proteins were hydrophilic proteins. Also, the vast majority of MedWRKYs’ instability index was greater than 40, indicating their structural instability. In addition, most MedWRKY proteins were localized in the nucleus, suggesting that they function mainly in the nucleus [26]. Phylogenetic analysis showed that 126 MedWRKY proteins could be classified into three groups, namely group I, II, and III. Group II could be divided into five subgroups, and this classification was consistent with previous studies [27]. Quantitatively, group II possessed the largest number of members with 80, accounting for approximately 63% of the total. Among the five subgroups of group II, II-b and II-c had the largest number of members. A similar distribution is found in other species such as drumstick [28], eucommia [29], and pineapple [30]. Multiple sequence alignment showed that MedWRKYs had a conservative WRKYGQK structure. In general, group I contained two WRKY domains, but a few MedWRKYs contained only one WRKY domain, such as MedWRKY9, MedWRKY 34, MedWRKY42, and so on. The same phenomenon was found in *Arabidopsis* [4], maize [31], and cassava [9]. At the same time, the WRKYGQK heptapeptide domain of some MedWRKY proteins was mutated, with R and Q mutated into K, forming WKKYGQK and WRKYGKK structures. Previous studies have shown that changes in the WRKYGQK motif in the WRKY domain may affect the interaction between WRKY genes and downstream target genes, implying that MedWRKYs are worthy of further study [32,33].

The gene structure and conserved motif were similar in the same group and subgroup, but there were differences among different groups. Different from other genes, *MedWRKY6* and *MedWRKY36* had no introns, which indicates that the *MedWRKY* genes may have exon and intron loss and gain events during the evolutionary process. This indicates the functional diversity of *MedWRKYs* [34]. In this study, we found that the promoter region of the *MedWRKY* genes contained multiple transcription factors binding sites associated with the transcription factors ZF-HD, bZIP, bHLH, etc. Different transcription factors have different regulatory roles. Studies have shown that the *Arabidopsis* bZIP gene *AtORG* inhibited leaf cell division, thereby affecting leaf enlargement [35]. Overexpression of *OsbHLH148* under drought conditions can improve plant tolerance [36]. In *Salvia miltiorrhiza*, *SmbZIP1* and *SmbZIP2* negatively regulated the biosynthesis of tanshinone and phenolic acids, respectively [37,38]. Promoter cis-acting element analysis can help predict the potential function of genes. The data showed that most of cis-acting elements were related to light-responsive elements and hormone-responsive elements, indicating that *MedWRKYs* might play an important role in plant growth and development and the hormonal regulation of plant biological processes. Through the collinear analysis of the WRKY gene family of *M. dodecandrum*, *E. grandis*, *A. thaliana*, and *O. sativa*, it was found that *M. dodecandrum* and *E. grandis* had the highest homology, which might be because both plants are dicotyledonous plants and belong to the order Myrtle. This implies that in the long-term evolution process, *M. dodecandrum* and *E. grandis* might have had some ancestors’ WRKY genes before differentiation [9].

The expression characteristics of plant genes are closely related to their functions. *AtWRKY75* had an important regulatory role in *Arabidopsis* root growth and development [39] and *AtWRKY12*, *AtWRKY13*, and *AtWRKY71* were involved in flower development in *Arabidopsis* [40,41]. SPF1 was the first WRKY protein identified, and it was mainly involved in sweet potato tuber development [5]. In addition, *OsWRKY11* was reported to be involved in flower development in rice [42], and *OsWRKY78* was hypothesized to be a regulator of stem elongation and seed development in rice [43]. In the present study, we examined the expression levels of 126 WRKY genes in different tissues of *M. dodecandrum*. All the genes had different expression levels in different tissues, and some *MedWRKY* genes (*MedWRKY14*, *35*, *41*, *50*, *60*, *61*, *65*, *114*, *119*, and *125*) were expressed at the highest levels in the root and ripe fruit tissue, suggesting their important roles in the growth and development of root and fruit tissue. Meanwhile, we also found that the vast majority of the highly expressed genes were concentrated in group II, indicating *MedWRKY* genes of group II may play more important roles in regulating the growth and development of plants. However, how exactly these genes are involved in the growth and development of *M. dodecandrum* needs to be further investigated.

## 4. Materials and Methods

### 4.1. Data Sources and Plant Materials

The plant materials were collected from the Soil and Water Conservation Garden of Fujian Agriculture and Forestry University. During the vegetative growth of *M. dodecandrum*, tender root, tender stem, and tender leaves were used as materials. During the flowering period, mature flowers were used as materials. Ripe fruits were collected during the fruiting period. All materials were immediately frozen in liquid nitrogen and stored at −80 °C for later analysis. The genomic data and transcriptome data of *M. dodecandrum* used in this study came from our previous work by Hao et al. [25].

### 4.2. Gene Identification and Physicochemical Property Analysis

Two methods were used to obtain the WRKY members using the *M. dodecandrum* genomic data. First, the Hidden Markov Model (HMM) file of the WRKY structural domain (PF03106) was downloaded from the Pfam protein family database (http://pfam.sanger.ac.uk/, accessed on 2 December 2022) and the WRKY gene was searched in the *M. dodecandrum* genome database, using the Hmmer search function in the HMMER 3.0 program (default parameters). Then, the protein sequences of the *A. thaliana* WRKY gene family were downloaded as seed files from the plant transcription factor website (http://planttfdb.gao-lab.org, accessed on 1 December 2022). The *M. dodecandrum* protein database was searched, using the BLAST method in the TBtools software (version 1.0987671, China), to obtain WRKY candidate members [44]. The two results were compared, and the common members were taken. The WRKY-conserved structural domains of the corresponding sequences were retrieved using the protein-conserved structural domain analysis tools NCBI CD-search and Pfam. Proteins with incomplete WRKY structural domains were excluded, and the remaining WRKY members were named MedWRKY.

The amino acid number, molecular weight, isoelectric point, and instability index of the MedWRKY proteins were analyzed using the Protein Paramter Calc tool in TBtools software (version 1.0987671, China). The subcellular prediction of MedWRKY proteins was performed through the online website WoLFPSORT (https://wolfpsort.hgc.jp, accessed on 7 December 2022).

### 4.3. Multiple Sequence Alignment and Phylogenetic Analysis

ClustalW in MEGA 5.0 was used to confirm the multiple sequence alignment of MedWRKYs, and the results were visualized using Jalview software (accessed on 12 March 2023). Based on the multiple sequence alignment, a phylogenetic tree was constructed using the maximum likelihood (ML) method in MEGA 5.0 with the bootstrap value set to 1000. The evolutionary tree was embellished using the ITOL (iTOL: Interactive Tree Of Life (embl.de)) online website. According to the phylogenetic tree, the MedWRKYs were classified into different groups with reference to the classification of AtWRKYs.

### 4.4. Gene Structure and Conserved Motif Analysis

The exon–intron structure of *MedWRKYs* was visualized and analyzed by TBtools software (version 1.0987671, China) [44]. Conserved motif prediction of MedWRKY sequences was performed using the MEME (http://meme-suite.org, accessed on 29 March 2023) website, with the number of motifs set to 10 and other parameters as default [45].

### 4.5. Promoter Sequences Analysis

A promoter sequence of 1500 bp upstream of the *MedWRKY* genes transcription start site was obtained from the *M. dodecandrum* genome [25]. Then, the promoter cis-acting elements were analyzed using the PlantPAN website (PlantPAN 4.0 (ncku.edu.tw), accessed on 1 September 2023) and Plant CARE website (http://bioinformatics.psb.ugent.be/webtools/plantcare/html/, accessed on 16 February 2023). On the PlantPAN website, *A. thaliana* was used as a reference species and the selected cis-acting element, with a score of 1. Finally, the results were visualized by TBtools [46].

### 4.6. Chromosome Localization and Collinearity Analysis

Based on the annotation information in the genome of *M. dodecandrum*, the location information of the WRKY chromosome was extracted, and a chromosome localization map was drawn. The gene duplication events were analyzed using MCScanX with default parameter settings [47]. The relevant data of *A. thaliana*, *E. grandis,* and *O. sativa* were downloaded from the NCBI (National Center for Biotechnology Information (nih.gov), accessed on 9 March 2023). TBtools was used to perform a species collinearity analysis and plot the covariance.

### 4.7. Expression Level Analysis

According to the transcriptome data of *M. dodecandrum* [25], we analyzed the expression levels of WRKY genes in different tissues. Hisat [48] and Stringtie2 [49] were used to align and assemble the transcriptome data of *M. dodecandrum.* Based on the FPKM value, TBtools was used to draw the heatmap.

### 4.8. RT-qPCR Analysis 

In order to further analyze the expression patterns of the WRKY gene family, a RT-qPCR experiment was conducted. Primer Premier 5.0 software was used to design primers. The total RNA of the root, stem, leaf, mature flower, and ripe fruit was extracted by the TIANGEN DP441 Reagent Kit (Tiangen, Beijing, China). A HiScript III 1st Strand cDNA Synthesis Kit (+ gDNA wiper; Vazyme, Nanjing, China) was used to reverse transcribe RNA to cDNA. Based on the Taq Pro Universal SYBR qPCR Master Mix kit (Vazyme, Nanjing, China), the ABI 7500 Real-Time System (Applied Biosystems, Foster City, CA, USA) was used to analyze the RT-qPCR. The *Actin* gene acted as an internal reference. The 2^−ΔΔCT^ method was employed to calculate the relative expression of the genes. The experimental template consisted of a 96-well plate, each of which established a 20 μL reaction system. Biological replicates and technical replicates were performed 3 times.

## 5. Conclusions

In this study, we performed a comprehensive analysis of the WRKY gene family in *M. dodecandrum*. We studied their physicochemical properties, phylogenetic construction, conserved motif characterization, gene structure, promoter sequences analysis, chromosomal localization, and covariance analysis. All the MedWRKY members were classified into three major groups, with similar gene structures and conserved motifs in the same group and subgroup. Phylogenetic and synteny analyses provided valuable clues for the evolutionary characterization of MedWRKYs. Moreover, the expression levels of *MedWRKYs* in different tissues were analyzed based on the transcriptome data. These results lay the foundation for subsequent functional studies of the *M. dodecandrum* WRKY gene family.

## Figures and Tables

**Figure 1 ijms-24-14904-f001:**
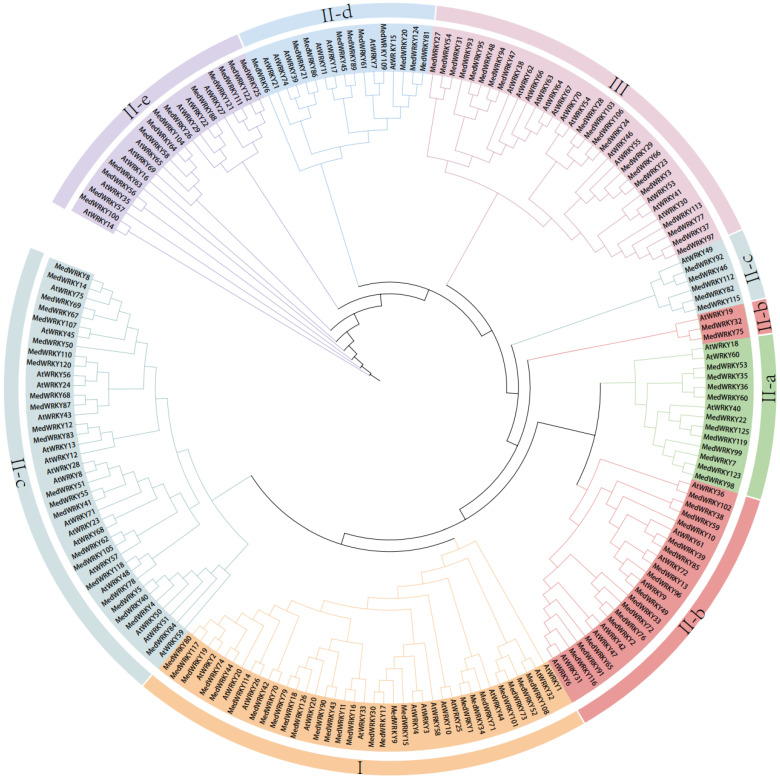
Phylogenetic tree of WRKY proteins from *M. dodecandrum* and *A. thaliana.* Note: “At” represents *A. thaliana*, and “Med” represents *M. dodecandrum*.

**Figure 2 ijms-24-14904-f002:**
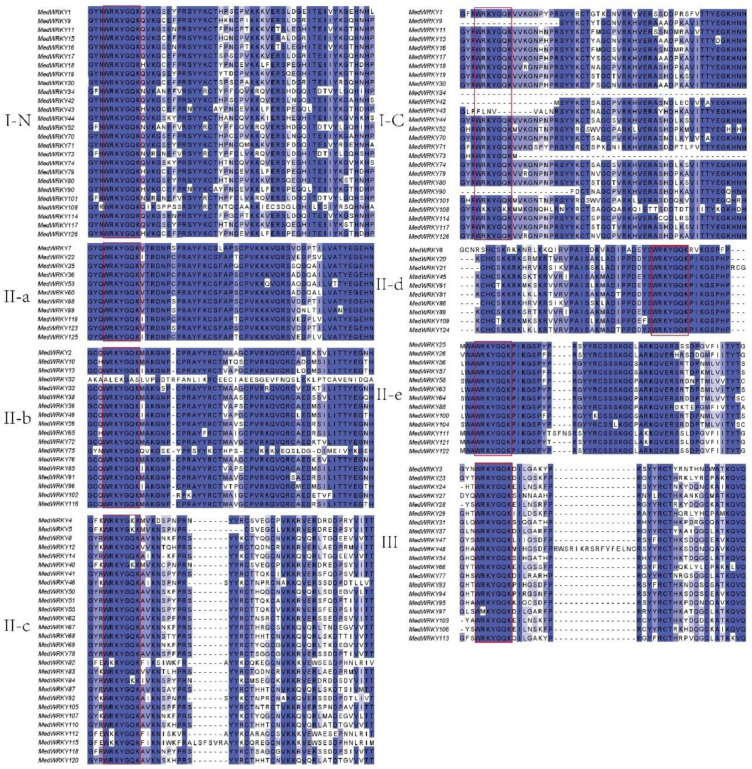
Multiple sequence alignment of MedWRKYs. Note: “N” and “C” represent the N-terminal and C-terminal WRKY domains of Group I, respectively. The conserved WRKY amino acid sequence is highlighted in red frames. The conserved WRKY amino acid sequence and zinc finger structure are indicated in dark blue.

**Figure 3 ijms-24-14904-f003:**
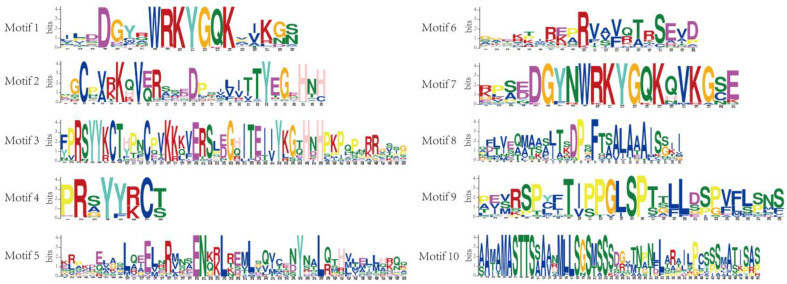
The sequences of MedWRKY proteins’ conserved motifs.

**Figure 4 ijms-24-14904-f004:**
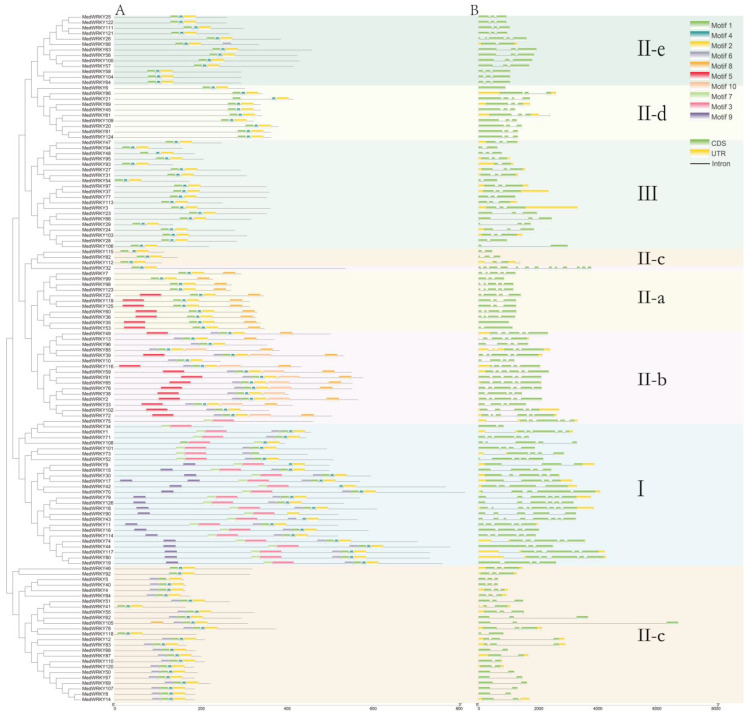
The motif distribution and gene structure analysis of MedWRKYs. Note: (**A**) conserved motifs of MedWRKYs. (**B**) Exon–intron structures of *MedWRKYs*.

**Figure 5 ijms-24-14904-f005:**
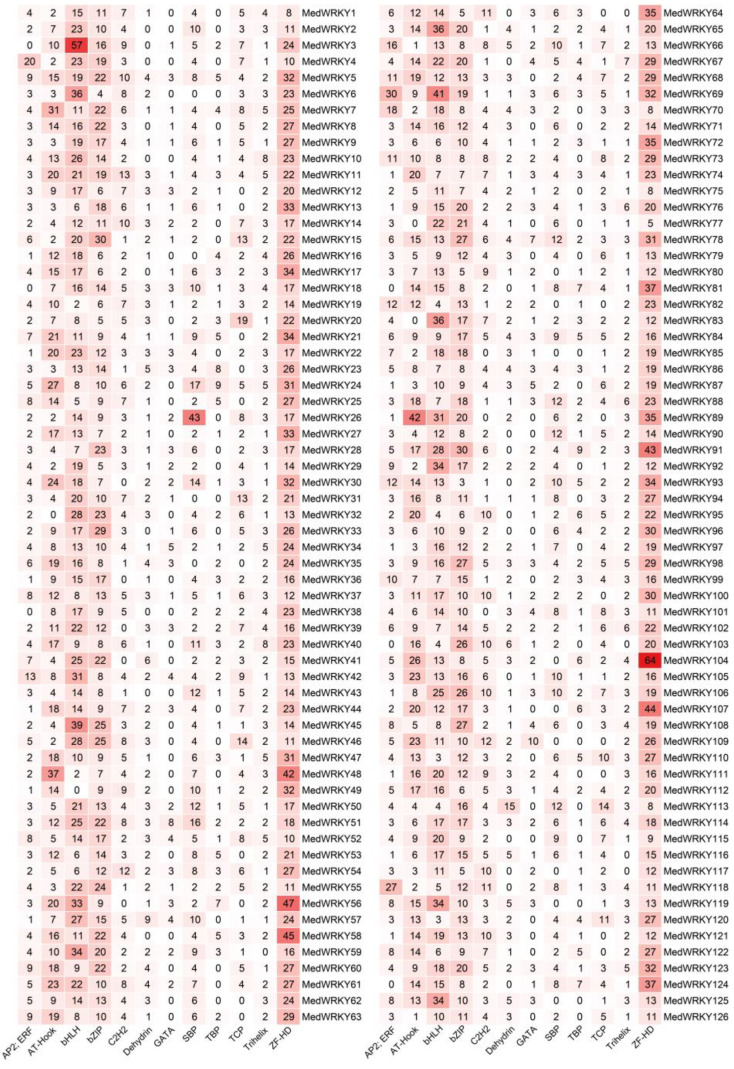
Promoter sequences analysis of *MedWRKYs*. Note: the numbers in the figure represent the number of transcription factor binding sites. The darker the color, the greater the number.

**Figure 6 ijms-24-14904-f006:**
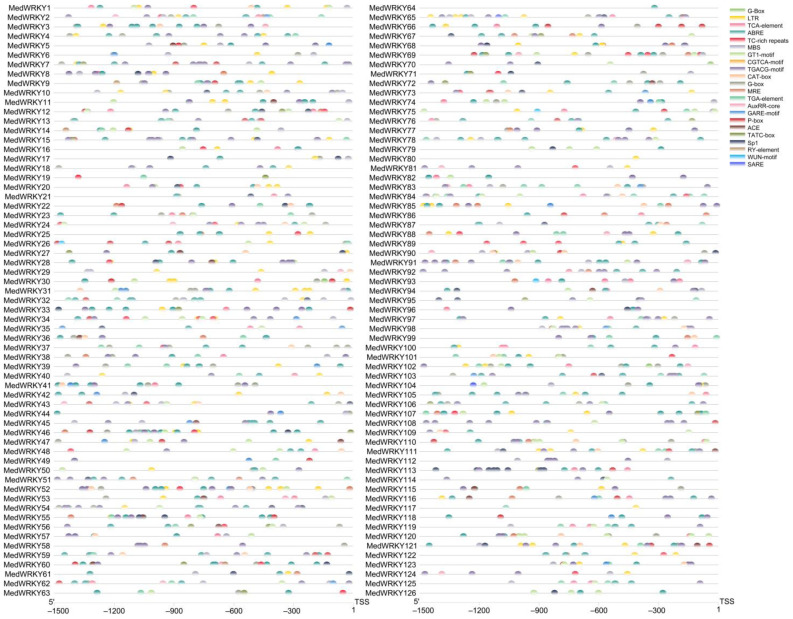
Analysis of cis-acting elements in the promoter regions of *MedWRKYs*. Note: different colored boxes represent different cis-acting elements. ”TSS”: transcription start site.

**Figure 7 ijms-24-14904-f007:**
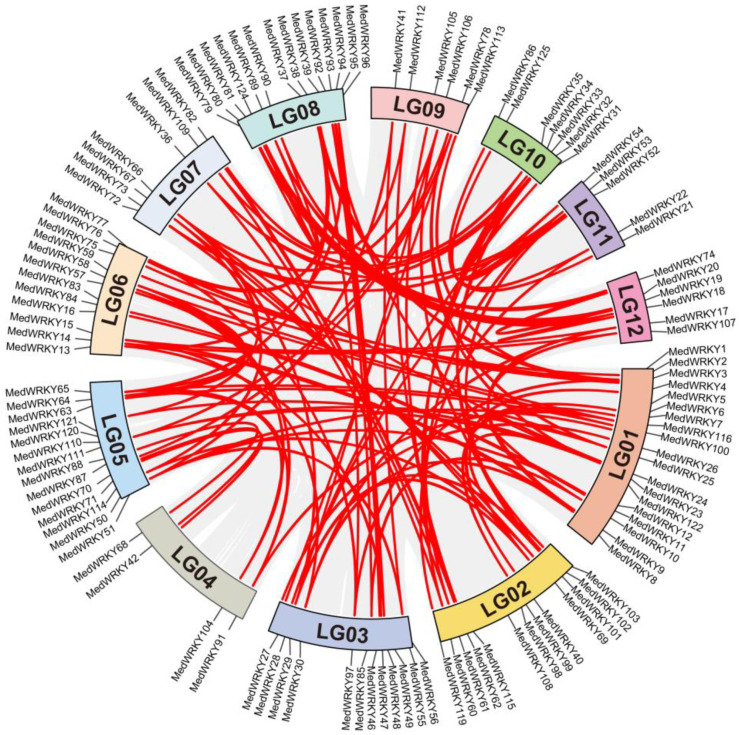
Chromosome localization of *MedWRKYs*. Note: the gray lines represent all collinear segments in the *M. dodecandrum* genome, and the red lines represent duplicate *MedWRKY* gene pairs. Rectangles of different colors indicate chromosomes.

**Figure 8 ijms-24-14904-f008:**
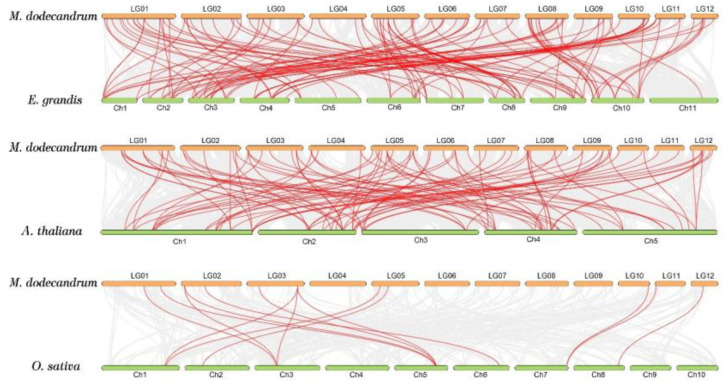
Synteny analysis of *MedWRKYs* between *M. dodecandrum* and three representative plant species. Note: the gray lines represent the collinear gene pair of *M. dodecandrum* and other species, and the red lines represent the collinear WRKY gene pairs.

**Figure 9 ijms-24-14904-f009:**
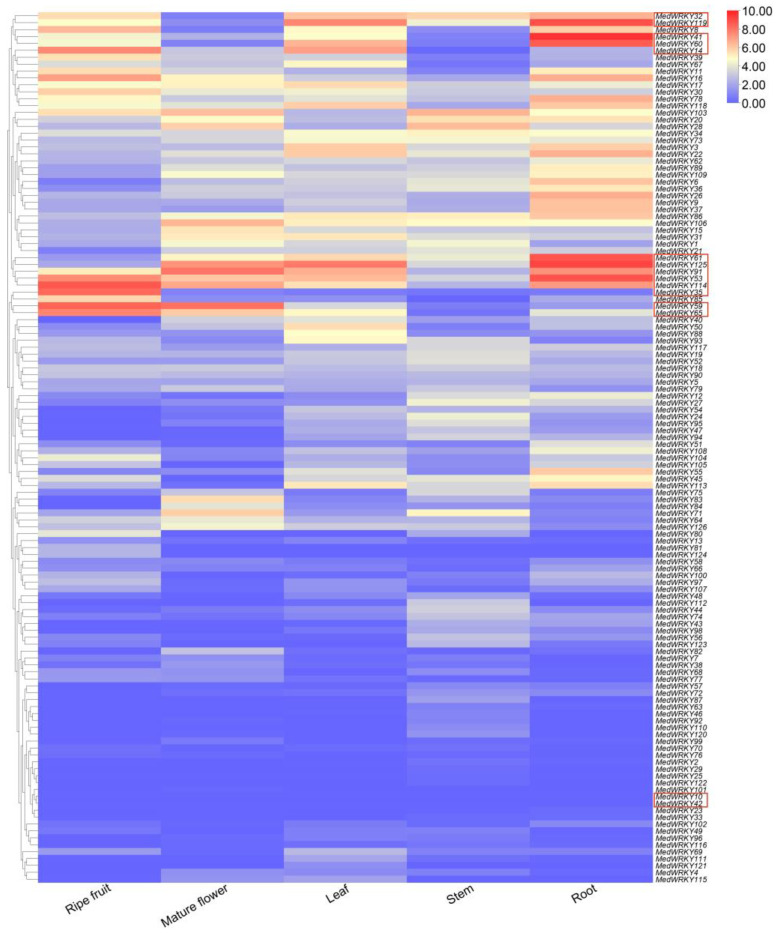
The expression level of *MedWRKYs* in different tissues. Note: different colors represent different FPKM values. Red and blue indicate high and low expression levels of *MedWRKYs*, respectively. Genes mentioned in the text are highlighted in red frames. ‘Ripe fruit’: the fruit is fully ripe and has a purplish-black outer skin.

**Figure 10 ijms-24-14904-f010:**
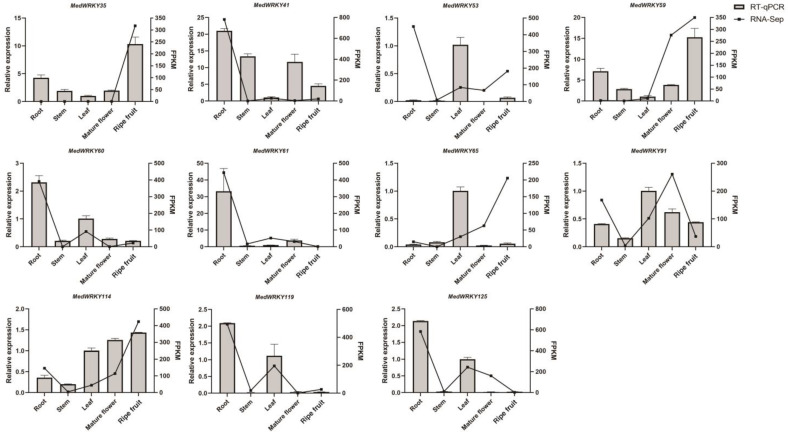
Expression profiles of different tissues of *MedWRKYs* by RT-qPCR. Note: bars represent the mean values of three technical replicates ± SE. The histograms represent RT-qPCR data, and line charts represent FPKM data.

## Data Availability

Not applicable.

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
