# Peer review of "Genome-Wide Identification and Analysis of WRKY Gene Family in *Melastoma dodecandrum"

_ijms, 2023, doi:10.3390/ijms241914904_

Round 1

Reviewer 1 Report

The manuscript entitled ‘Genome-wide Identification and Analysis of WRKY Gene Family in Melastoma dodecandrum’ by Tang et al. aimed to characterize the WRKY family in M. dodecandrum. The authors have identified that MedWRKY may be crucial in controlling plant growth. The authors have tried their best to verify their findings.

However, I have the following comments to improve the standard of the manuscript:

1.      The Introduction needs revision particularly on the regulatory role of WRKY transcription factor in response to growth, development, and abiotic and biotic stresses.

2.      Under the section: 4.3. Analysis of Gene structure, Conserved Motif and Cis-Regulatory Elements, line 311-318, the authors have mentioned, “To determine the type of cis-regulatory elements, it was analyzed using the online tool PlantCARE (http://bioinformatics.psb.ugent.be/webtools/plantcare/html/) by extracting the sequence of 1500 bp upstream of  the MedWRKY gene [22].” Please clearly mention regarding the extraction of 1500 bp promoter sequences.  From which site (point) did you extract the promoter sequences?  The nucleotide numbering in the Figure 5 for the promoter cis-element analysis and 3’ position are not correct. Please revise it.

The authors have used Plant CARE database for promoter cis-element analysis. However, this database is very old and does not have up to date information. Hence, you are certainly going to miss many important and new cis-elements. The best option is to use the MATCH program in TRANSFAC (geneXplain) which you need to pay for the subscription. However, the authors can also use PlantPAN and PLACE database if you do not have access to TRANSFAC.

3.      The authors have mentioned regarding M. dodecandrum “big fruit” throughout the manuscript. What do you mean by big fruit? Is it matured fruit? Please clarify.

4.         In the discussion, please revise the promoter cis-element analysis part based on your new analysis.

Please refine the language and grammar. 

Author Response

Response to Reviewer 1

Dera reviewers,

On behalf of co-authors, we thank you very much for giving us an opportunity to revise our manuscript. We appreciate reviewer very much for your useful and constructive comments and suggestions on our manuscript entitled “Genome-wide Identification and Analysis of WRKY Gene Family in Melastoma dodecandrum”. In this revised version, we have addressed the concerns of the reviewers. An item-by-item response to your comments is enclosed, and the revision was marked in red fonts in the manuscript. The detailed corrections are listed below.

Reviewer 1

  1. Comment: The Introduction needs revision particularly on the regulatory role of WRKY transcription factor in response to growth, development, and abiotic and biotic stresses.
  2. Reply: Thank you for your suggestion. Regarding the content of this part, we revised it in Line 49-Line 61.
  3. Comment: Under the section: 4.3. Analysis of Gene structure, Conserved Motif and Cis-Regulatory Elements, line 311-318, the authors have mentioned, “To determine the type of cis-regulatory elements, it was analyzed using the online tool PlantCARE (http://bioinformatics.psb.ugent.be/webtools/plantcare/html/) by extracting the sequence of 1500 bp upstream of  the MedWRKY gene [22].” Please clearly mention regarding the extraction of 1500 bp promoter sequences.  From which site (point) did you extract the promoter sequences?  The nucleotide numbering in the Figure 5 for the promoter cis-element analysis and 3’ position are not correct. Please revise it.

The authors have used Plant CARE database for promoter cis-element analysis. However, this database is very old and does not have up to date information. Hence, you are certainly going to miss many important and new cis-elements. The best option is to use the MATCH program in TRANSFAC (geneXplain) which you need to pay for the subscription. However, the authors can also use PlantPAN and PLACE database if you do not have access to TRANSFAC.

  1. Reply: Regarding the extraction of promoter sequences, we have supplemented them in Line 325 and the relevant information such as site in the attachment file(Table S2). There are errors in Figure 5, and we analyzed and re-plotted them (Line 163).

The analysis using the Plant CARE database was based on analysis methods from other essays. We also tried to analyze it with PlantPAN and PLACE, but due to unfamiliarity with these two methods and the large number of genes, we are sorry not to analyze with this method.

  1. Comment: The authors have mentioned regarding M. dodecandrum“big fruit” throughout the manuscript. What do you mean by big fruit? Is it matured fruit? Please clarify.
  2. Reply: “Big fruit”: the fruit is fully ripe and has a purplish-black outer skin. We provided additional description at Line 206.
  3. Comment: In the discussion, please revise the promoter cis-element analysis part based on your new analysis.
  4. Reply: Based on the new analysis, we have revised the relevant content in the discussion (Line 256).

Thank you very much for your attention and time. Look forward to hearing from you.

Yours sincerely,

Dong-Hui Peng

20 August 2023

Fujian Agriculture and Forestry University

Reviewer 2 Report

The manuscript by Tang et al. presents an analysis of the Melastoma dodecandrum WRKY genes. In the first part of the paper, the authors analyzed the previously obtained genomic data, predicted 126 WRKY genes and classified them into three major groups. The authors analyzed the exon-intron structure and cis-elements of the WRKY genes and predicted physicochemical properties of the corresponding proteins, analyzed the phylogeny and conserved motifs. Next, authors presented the transcriptomic and qPCR expression data of WRKY genes, however, the origin of these data remains unclear, as there is no any information about the plant material used for experiments.

General comments:

1)    In many cases, the authors describe protein properties as the properties of genes. For example, “Results”, line 71: “Physicochemical Property Analysis of MedWRKY Gene”. Corresponding sentence in the “Discussion”, line 229: “Physicochemical characterization showed that MedWRKY genes were hydrophilic proteins”.

2)    To my mind, the analysis of gene structure and predicted cis-elements should be described separately from the analysis of the TFs encoded be these genes.

The cis-elements of WRKY genes are associated with the regulation of these genes, while the sequences of WRKY TFs are associated with the regulation of other genes by these TFs.

3)    Please use Italics for gene names throughout the paper.

4)    The authors provide no information about the origin of the transcriptomic and qPCR data. There should be a detailed description of the dataset, sample collection, sequencing and data processing. In a case if the authors used the previously published data, the reference is required. In the description of the qPCR data, the authors also do not say a word about the plant material used. Did the authors use the same biological samples for transcriptomics and qPCR?

5)    Lines 134, 143, 173, etc: “MedWRKY gene”.  Should be “MedWRKY genes”.

Other comments: 

6)    Line 14: “WRKY is one of the largest transcription regulatory factors”. Transcription factor families?

7)    Line 26: “the expression of these genes was much higher in the growth region compared to the mature group”. The meaning of this sentence is unclear.

8)    The image resolution in Figure 4 should be improved.

9)    Line 155: “These results revealed that the transcriptional regulation of MedWRKY…” These results suggested…<>.

10)  Line 189: “As shown in Fig.8, each MedWRKY gene was expressed in at least 1 tissue, except MedWRKY10 and MedWRKY42”. It is very hard to find the mentioned genes in the Figure, which presents all the dataset. I suggest the authors to improve presentation of these data, for example, to select the mentioned genes, make the zoomed insets, etc.

11)  Line 188: “The results showed that the MedWRKY gene was differentially expressed…” Line 197: “The WRKY genes, which were not significantly differentially expressed in other tissues…”

The authors mention the differential expression analysis, but I think it is a wrong term here.

12)  Line 202, Figure 8 legend: What values are shown in the heat map? FPKM? As I have already mentioned, this analysis needs the detailed methodological description.

13)  Line 215, Figure 9 legend: This figure also lacks the detailed description. The authors should decipher the units in which expression was measured, specify the meaning of error bars and provide the number of biological and technical replicates for each tissue.

14)  Line 328: “..we analyzed the expression patterns of WRKY genes in different organizations.” In different tissues?

15)  Line 332: As I have already mentioned, there is no information about the growth conditions, plant material collection, RNA extraction, etc.

Author Response

Response to Reviewer 2

Dear reviewers,

On behalf of co-authors, we thank you very much for giving us an opportunity to revise our manuscript. We appreciate reviewer very much for your useful and constructive comments and suggestions on our manuscript entitled “Genome-wide Identification and Analysis of WRKY Gene Family in Melastoma dodecandrum”. In this revised version, we have addressed the concerns of the reviewers. An item-by-item response to your comments is enclosed, and the revision was marked in red fonts in the manuscript. The detailed corrections are listed below.

Reviewer 2

General comments:

1. Comment: In many cases, the authors describe protein properties as the properties of genes. For example, “Results”, line 71: “Physicochemical Property Analysis of MedWRKY Gene”. Corresponding sentence in the “Discussion”, line 229: “Physicochemical characterization showed that MedWRKY genes were hydrophilic proteins”.

1. Reply:Thanks for your careful check. These two errors have been corrected in Line 74 and Line 233.2.

2. Comment: To my mind, the analysis of gene structure and predicted cis-elements should be described separately from the analysis of the TFs encoded be these genes.

The cis-elements of WRKY genes are associated with the regulation of these genes, while the sequences of WRKY TFs are associated with the regulation of other genes by these TFs.

2. Reply: Thank you for your suggestion. We’re sorry to say we don't fully understand your suggestion.

3. Comment: Please use Italics for gene names throughout the paper.

3. Reply: Thanks for your reminder. We examined the paper and corrected the genes that were not italics.

4. Comment: The authors provide no information about the origin of the transcriptomic and qPCR data. There should be a detailed description of the dataset, sample collection, sequencing and data processing. In a case if the authors used the previously published data, the reference is required. In the description of the qPCR data, the authors also do not say a word about the plant material used. Did the authors use the same biological samples for transcriptomics and qPCR?

4. Reply: We've added “Data Sources and Plant Materials” to the “Materials & Methods” section(Line 289). The relevant data is mainly derived from Hao (Line 293). In the “RT -qPCR Analysis” part, RNA extraction and RT-qPCR data analysis related content were added (Line 345). And we used the same biological samples for transcriptomics and qPCR.

5. Comment: Lines 134, 143, 173, etc: “MedWRKY gene”.  Should be “MedWRKY genes”.

5. Reply: Thanks for your carefulness. We have changed all the “MedWRKY gene” in the article to “MedWRKY genes”.

Other comments:

6. Comment:Line 14: “WRKY is one of the largest transcription regulatory factors”. Transcription factor families?

6. Reply: Thanks for your reminder. We have modified it (Line 14).

7. Comment:Line 26: “the expression of these genes was much higher in the growth region compared to the mature group”. The meaning of this sentence is unclear.

7. Reply: According to the FPKM value, we count the values of the growth region and the mature region, and the results show that the expression of the growth region is higher than that of the mature region. And We added FPKM-related content to the paper (Table S3).

8. Comment:The image resolution in Figure 4 should be improved.

8.Reply: Thanks for your reminder. We have increased the resolution of Figure 4.

9. Comment:Line 155: “These results revealed that the transcriptional regulation of MedWRKY…” These results suggested…<>.

9. Reply: We have corrected it toThese results suggested that the transcriptional regulation of MedWRKY…” (Line 158).

10. Comment:Line 189: “As shown in Fig.8, each MedWRKY gene was expressed in at least 1 tissue, except MedWRKY10 and MedWRKY42”. It is very hard to find the mentioned genes in the Figure, which presents all the dataset. I suggest the authors to improve presentation of these data, for example, to select the mentioned genes, make the zoomed insets, etc.

10. Reply: Thanks for your reminder. For the genes mentioned in the paper, we highlight them in the figure with a red frame (Line 203).

11. Comment:Line 188: “The results showed that the MedWRKY gene was differentially expressed…” Line 197: “The WRKY genes, which were not significantly differentially expressed in other tissues…”

The authors mention the differential expression analysis, but I think it is a wrong term here.

11. Reply: We have modified these two sentences to make their content clearer in expression (Line 190, Line 199).

12. Comment:Line 202, Figure 8 legend: What values are shown in the heat map? FPKM? As I have already mentioned, this analysis needs the detailed methodological description.

12. Reply: Yes. The FPKM values were shown in the heat map. And we added methodological description in the paper (Line 340).

13. Comment:Line 215, Figure 9 legend: This figure also lacks the detailed description. The authors should decipher the units in which expression was measured, specify the meaning of error bars and provide the number of biological and technical replicates for each tissue.

13. Reply: We have added a detailed description of Figure 9 (Line 218). The number of biological and technical replicates was added in Line 354.

14. Comment:Line 328: “..we analyzed the expression patterns of WRKY genes in different organizations.” In different tissues?

14. Reply: We have modified this sentence to “we analyzed the expression levels of WRKY genes in different tissues” (Line 339).

15. Comment:Line 332: As I have already mentioned, there is no information about the growth conditions, plant material collection, RNA extraction, etc.

15. Reply: We've added “Data Sources and Plant Materials” to the “Materials & Methods” section(Line 289).In the “RT -qPCR Analysis” part, RNA extraction and RT-qPCR data analysis-related content were added (Line 345).

Thank you very much for your attention and time. Look forward to hearing from you.

Yours sincerely,

Dong-Hui Peng

20 August 2023

Fujian Agriculture and Forestry University

Round 2

Reviewer 1 Report

Thank you authors for revising the manuscript.

However, I still have the following major comments for the authors, as they have not correctly revised:

1.      In Materials and Methods, 4.5. Cis-Acting Elements Analysis, line 326-330, authors have mentioned, “Tbtools was used to extract the upstream 1500 bp sequence of the WRKY gene family from the genomic data of M. dodecandrum. Then the sequences were uploaded to the online  website PlantCARE(http://bioinformatics.psb.ugent.be/webtools/plantcare/html/) to predict the  cis-acting elements, and the result was visualized by TBtools [47].” You did not indicate from which point (Site) you extracted the upstream 1500 bp promoter sequences. It is important to indicate regarding the exact location of the promoter sequences. Please provide the information.

The authors have used Plant CARE database for promoter cis-element analysis. However, this database is very old and does not have the up to date information. Hence, you are certainly going to miss many important and new cis-elements. The best option is to use TRANSFAC (geneXplain). However, the authors can also use PlantPAN and PLACE database if you do not have access to TRANSFAC. It is not difficult to use those databases. Hence, you can try using those databases to improve your manuscript.

2.      Since the authors have fundamental problem in understanding the promoter sequences, the nucleotide numbering and 3’ position in the Figure 5 for the promoter cis-element analysis is not correct. Please correct it.

3.      In the Materials and Methods, Results and Discussion, please revise the promoter cis-element analysis part based on your new analysis.

4.      Instead of mentioning ‘big fruit’, it is better to write ‘Ripe fruit’. Big fruit does not sound appropriate.

Minor editing is necessary.

Author Response

Dear reviewers,

On behalf of co-authors, we thank you very much for giving us an opportunity to revise our manuscript. We appreciate reviewers very much for your useful and constructive comments and suggestions on our manuscript entitled “Genome-wide Identification and Analysis of WRKY Gene Family in Melastoma dodecandrum”. In this revised version, we have addressed the concerns of the reviewers. An item-by-item response to the reviewers' comments is enclosed, and the revision was marked in blue fonts in the manuscript. The detailed corrections are listed below.

Reviewer 1

1.Comment: In Materials and Methods, 4.5. Cis-Acting Elements Analysis, line 326-330, authors have mentioned, “Tbtools was used to extract the upstream 1500 bp sequence of the WRKY gene family from the genomic data of M. dodecandrum. Then the sequences were uploaded to the online  website PlantCARE(http://bioinformatics.psb.ugent.be/webtools/plantcare/html/) to predict the  cis-acting elements, and the result was visualized by TBtools [47].” You did not indicate from which point (Site) you extracted the upstream 1500 bp promoter sequences. It is important to indicate regarding the exact location of the promoter sequences. Please provide the information.

The authors have used Plant CARE database for promoter cis-element analysis. However, this database is very old and does not have the up to date information. Hence, you are certainly going to miss many important and new cis-elements. The best option is to use TRANSFAC (geneXplain). However, the authors can also use PlantPAN and PLACE database if you do not have access to TRANSFAC. It is not difficult to use those databases. Hence, you can try using those databases to improve your manuscript.

1.Reply: Thank you for your suggestion. We have already performed the analysis using PlantPAN. And we have added a description of the extraction site in the text. (Line 321)

2.Comment:Since the authors have fundamental problem in understanding the promoter sequences, the nucleotide numbering and 3’ position in the Figure 5 for the promoter cis-element analysis is not correct. Please correct it.

2.Reply: Because the analysis was re-analyzed using PlantPAN, and the number of results obtained was large, so it was shown with a heat map (Line 155).

3.Comment:In the Materials and Methods, Results and Discussion, please revise the promoter cis-element analysis part based on your new analysis.

3.Reply: Thanks for your reminder. We have corrected it in the corresponding places in the text. (Line 21, Line 143, Line 249 and Line 319)

4.Comment:Instead of mentioning ‘big fruit’, it is better to write ‘Ripe fruit’. Big fruit does not sound appropriate.

4.Reply: Thank you for your suggestion. We have changed the article that refers to ‘big fruit’ to ‘ripe fruit’.

Reviewer 2 Report

The manuscript by Tang et al. has been revised, but the main points were not fully addressed by the authors.

1)    I still do not understand whether the biological material was obtained by this group earlier or whether it was collected specifically for this study. In the section “Data Sources and Plant Materials” the authors write: “The genomic, transcriptome and other data of M. dodecandrum used in this study came from Hao [25].”

In the Cover letter, the authors say:  “And we used the same biological samples for transcriptomics and qPCR”. This phrase should be included in the manuscript.

Overall, the authors must specify, which materials/data were taken from the earlier published sources, and which were obtained in the current study.

Section 4.1. should also include the details regarding the biological material. At what stage of development was each tissue collected?

2)    In my comment regarding the sentence in Line 26 (“the expression level of these genes was much higher in the growth region compared to the mature group”), I wanted to clarify what is meant by the “growth region” and “mature group”. The authors should specify this in the text of the manuscript.

3)    Line 189: “Based on the M. dodecandrum transcriptome data…” There is no reference to the previous publication here.

Line 340: “According to the transcriptome data of M.dodecandrum, we analyzed the expression levels of WRKY genes in different tissues.” No reference either.

4)    Figure 8: The authors highlighted the genes mentioned in the text with red frames, but did not write about it in the legend to the figure.

5)    In my previous review, I wrote: “In many cases, the authors describe protein properties as the properties of genes”. The authors still did not fully address this point. Please use the terms "protein" and "gene" correctly in the text and Figure legends.

6)    In this regard, MedWRKY (and other names) should be italicized when referring to genes and not italicized when referring to proteins (both in the text and in the figures).

Author Response

Response to reviewers

Dear reviewers,

On behalf of co-authors, we thank you very much for giving us an opportunity to revise our manuscript. We appreciate reviewers very much for your useful and constructive comments and suggestions on our manuscript entitled “Genome-wide Identification and Analysis of WRKY Gene Family in Melastoma dodecandrum”. In this revised version, we have addressed the concerns of the reviewers. An item-by-item response to the reviewers' comments is enclosed, and the revision was marked in blue fonts in the manuscript. The detailed corrections are listed below.

Reviewer 2

1.Comment:I still do not understand whether the biological material was obtained by this group earlier or whether it was collected specifically for this study. In the section “Data Sources and Plant Materials” the authors write: “The genomic, transcriptome and other data of M. dodecandrumused in this study came from Hao [25].”

In the Cover letter, the authors say:  “And we used the same biological samples for transcriptomics and qPCR”. This phrase should be included in the manuscript.

Overall, the authors must specify, which materials/data were taken from the earlier published sources, and which were obtained in the current study.

Section 4.1. should also include the details regarding the biological material. At what stage of development was each tissue collected?

1.Reply: Thank you for your suggestion. The plant material used in this study came from the previous collection of the research group. We have added the corresponding description in “Data Sources and Plant Materials”. (Line 280)

2.Comment:In my comment regarding the sentence in Line 26 (“the expression level of these genes was much higher in the growth region compared to the mature group”), I wanted to clarify what is meant by the “growth region” and “mature group”. The authors should specify this in the text of the manuscript.

2.Reply: Thank you for your suggestion. The “mature group” was changed to the “mature tissue”. The “growth region” refers to the root, stem and leaf. The “mature tissue” refers to the mature flower and ripe fruit. (Line 26)

3.Comment:Line 189: “Based on the M. dodecandrumtranscriptome data…” There is no reference to the previous publication here.

Line 340: “According to the transcriptome data of M.dodecandrum, we analyzed the expression levels of WRKY genes in different tissues.” No reference either.

3.Reply: Thanks for your reminder. We have added references in Line 180 and Line 334.

4.Comment:Figure 8: The authors highlighted the genes mentioned in the text with red frames, but did not write about it in the legend to the figure.

4.Reply: Thanks for your careful check. We've added a description in Line 198.

5.Comment:In my previous review, I wrote: “In many cases, the authors describe protein properties as the properties of genes”. The authors still did not fully address this point. Please use the terms "protein" and "gene" correctly in the text and Figure legends.

5.Reply: Thanks for your reminder. We have already made relevant corrections in the article.

6.Comment:In this regard, MedWRKY(and other names) should be italicized when referring to genes and not italicized when referring to proteins (both in the text and in the figures).

6.Reply: Thanks for your reminder. We have already made relevant corrections in the article.

Round 3

Reviewer 1 Report

Thank you authors for the revision.

However, I have the following major comment for the authors:

1.      In the Materials and Methods, under section 4.5. Cis-Acting Elements Promoter Sequences Analysis, line 375-377, the authors mentioned, “Based on the genomic data of M. dodecandrum [25], TBtools was used to extract the DNA sequence of 1500 bp upstream of the WRKY gene family from  the genomic data of M. dodecandrum initiation codon (ATG)."

The authors have taken 1500 bp upstream of the initiation codon (ATG) site, which is Translation Initiation Site (TIS). The sequences they have taken could be a part of Exon 1 and a part of promoter or only Exon 1 depending on the gene length. So, the cis-element detected in those regions are not correct, as the sequences are not right promoter sequences. Sometime, some distal cis-elements are present in the downstream of promoter sequence, but majority are present in the upstream part of promoter sequence. Hence, please extract the promoter sequences from the upstream of TSS (Transcription Start Site) and redo the cis-element enrichment analysis.

2.      Please revise the method, results and discussion part based on your new promoter cis-element analysis.

3.      Why the number of binding sites are so high? You need to filter out the number of binding sites based on their scores.

4.      Line 173-175, the authors have mentioned, “Analysis of the promoter sequences of 1500 bp upstream showed that many transcription factors related to plant growth and development, stress response, and secondary  metabolite synthesis were present in the WRKY gene family.” Actually transcription factors are not present in the WRKY gene family, instead the WRKY genes have binding sites for different TFs. Please revise such kind of sentences throughout the manuscript.

5.      In the results, line 82-83, the authors mentioned, “After searching, a total of 126 MedWRKY proteins were finally obtained, which were named MedWRKY1~MedWRKY126.”  Should it be 126 MedWRKY proteins or genes?

6.      In the Discussion part, line 300-307, the authors mentioned, “Promoter sequence analysis showed that the regulatory role of MedWRKY genes may be affected by a variety of transcription factors. For example, bHLH, bZIP, and other transcription factors were not only involved in plant growth and development but also played  a key role in plant response to stress and secondary metabolism. Studies have shown that Arabidopsis bZIP gene AtORG inhibited leaf cell division, thereby affecting leaf enlargement [35]. Overexpression of OsbHLH148 under drought conditions can improve plant tolerance [36]. In Salvia miltiorrhiza, SmbZIP1 and SmbZIP2 negatively regulated the biosynthesis of tanshinone and phenolic acids, respectively [37-38].” This part does not provide any information regarding the presence of different cis-elements in the promoters of WRKY genes. Please revise it. MedWRKY genes have binding sites for different TFs which shows that WRKY genes could be involved in the regulation of growth, development, different abiotic and biotic stresses etc.

7.      Please check and italicize the genes.

8.      Please refine the language.

Please refine the language.

Author Response

Response to reviewers

Dear reviewers,

On behalf of co-authors, we thank you very much for giving us an opportunity to revise our manuscript. We appreciate reviewers very much for your useful and constructive comments and suggestions on our manuscript entitled “Genome-wide Identification and Analysis of WRKY Gene Family in Melastoma dodecandrum”. In this revised version, we have addressed the concerns of the reviewers. An item-by-item response to the reviewers' comments is enclosed, and the revision was marked in red fonts in the manuscript. The detailed corrections are listed below.

1.Comment: In the Materials and Methods, under section 4.5. Cis-Acting Elements Promoter Sequences Analysis, line 375-377, the authors mentioned, “Based on the genomic data of M. dodecandrum[25], TBtools was used to extract the DNA sequence of 1500 bp upstream of the WRKY gene family from  the genomic data of M. dodecandruminitiation codon (ATG)."

The authors have taken 1500 bp upstream of the initiation codon (ATG) site, which is Translation Initiation Site (TIS). The sequences they have taken could be a part of Exon 1 and a part of promoter or only Exon 1 depending on the gene length. So, the cis-element detected in those regions are not correct, as the sequences are not right promoter sequences. Sometime, some distal cis-elements are present in the downstream of promoter sequence, but majority are present in the upstream part of promoter sequence. Hence, please extract the promoter sequences from the upstream of TSS (Transcription Start Site) and redo the cis-element enrichment analysis

1.Reply: Thank you for your suggestion. We have made relevant modifications in Line 336.

2.Comment: Please revise the method, results and discussion part based on your new promoter cis-element analysis.

2.Reply: Thank you for your reminder. We have already made relevant corrections to the article. (Line 142, Line 260, Line 335)

3.Comment:Why the number of binding sites are so high? You need to filter out the number of binding sites based on their scores.

3.Reply: Thank you for your suggestion. Based on their scores, we selected transcription factor binding sites with a score of 1. And we re-analysed and mapped. (Line 339)

4.Comment:Line 173-175, the authors have mentioned, “Analysis of the promoter sequences of 1500 bp upstream showed that many transcription factors related to plant growth and development, stress response, and secondary metabolite synthesis were present in the WRKY gene family.” Actually transcription factors are not present in the WRKY gene family, instead the WRKY genes have binding sites for different TFs. Please revise such kind of sentences throughout the manuscript.

4.Reply: Thank you for your reminder. We have already made relevant corrections to the article. (Line 21、Line143、Line 260)

5.Comment:In the results, line 82-83, the authors mentioned, “After searching, a total of 126 MedWRKY proteins were finally obtained, which were named MedWRKY1~MedWRKY126.”  Should it be 126 MedWRKY proteins or genes.

5.Reply: Thank you for your reminder. Here, it refers to proteins. The revised sentence is “After searching, 126 WRKY members were finally obtained, and the MedWRKYs were named MedWRKY1~MedWRKY126.” (Line 75)

6.Comment: In the Discussion part, line 300-307, the authors mentioned, “Promoter sequence analysis showed that the regulatory role of MedWRKY genes may be affected by a variety of transcription factors. For example, bHLH, bZIP, and other transcription factors were not only involved in plant growth and development but also played a key role in plant response to stress and secondary metabolism. Studies have shown that Arabidopsis bZIP gene AtORG inhibited leaf cell division, thereby affecting leaf enlargement [35]. Overexpression of OsbHLH148 under drought conditions can improve plant tolerance [36]. In Salvia miltiorrhizaSmbZIP1 and SmbZIP2 negatively regulated the biosynthesis of tanshinone and phenolic acids, respectively [37-38].” This part does not provide any information regarding the presence of different cis-elements in the promoters of WRKY genes. Please revise it. MedWRKY genes have binding sites for different TFs which shows that WRKY genes could be involved in the regulation of growth, development, different abiotic and biotic stresses etc.

6.Reply: Thank you for your suggestion. We have added a description of this in Line 266.

7.Comment: Please check and italicize the genes.

7.Reply: Thank you for your reminder. We've made changes to the content.

8.Comment: Please refine the language.

8.Reply: Thank you for your suggestion. We've made changes to the content.

Reviewer 2 Report

During the revision, the authors addressed many comments, but some issues still remain.

1)    After the revision, it became evident that the roots, leaves and stems are attributed to “growth regions”, while flowers and ripe fruits – to the “mature tissues”.

In this regard, I see the major issue: the conclusion that “the expression level of <MedWRKY> genes was much higher in the growth regions (root, stem, leaf) compared to the mature tissues (mature flower, ripe fruit), indicating that MedWRKYs may be crucial in controlling plant growth” does not correspond to the description of the MedWRKY gene expression presented in the section 2.6 (“MedWRKY14, MedWRKY35, MedWRKY65, and MedWRKY114 were most highly expressed in ripe fruit and low in other tissues. MedWRKY32 was moderately expressed in stem, leave, and ripe fruit, and low in mature flower. MedWRKY59 was highly expressed in mature flower and ripe fruit, and low expression in other tissues. MedWRKY91 was highly expressed in root, leave and mature flower, and low expression was seen in stem and ripe fruit.”) I am not sure if it can be inferred from this text that WRKY expression is much higher in root, stem and leaves compared to flowers and fruits. Figures 8 and 9 also do not reveal this tendency.

2)    Line 321: “most MedWRKY genes were mainly expressed in root”. This conclusion is also not supported by the data.

3)    To my mind, a more relevant conclusion is presented in line 414: “Based on the transcriptome data and RT-qPCR data, we hypothesize that MedWRKY transcription factors play important roles in different tissues and different developmental periods of M.dodecandrum”.

However, here “transcription factors” should be replaced by “genes” since the authors analyzed the levels of MedWRKY transcripts, not proteins. The mRNA and protein expression patterns may significantly differ due to post-transcriptional regulation.

4)    Line 226: “The WRKY genes, which were not expressed at significant levels in other tissues, were likely to play a more general role in M.dodecandrum.” What “other tissues” did the authors mean?

5)    “The plant material used in this study came from the previous collection of the research group.” Please include a reference here (ref. 25?)

6)    In Figures 8 and 9: some red horizontal lines appear in the middle of the figures.

7)    Figure 8, page 231, in the bottom: “big fruit” instead of “ripe fruit”. Gene names in the figure should be in Italics.

Author Response

Response to reviewers

Dear reviewers,

On behalf of co-authors, we thank you very much for giving us an opportunity to revise our manuscript. We appreciate reviewers very much for your useful and constructive comments and suggestions on our manuscript entitled “Genome-wide Identification and Analysis of WRKY Gene Family in Melastoma dodecandrum”. In this revised version, we have addressed the concerns of the reviewers. An item-by-item response to the reviewers' comments is enclosed, and the revision was marked in red fonts in the manuscript. The detailed corrections are listed below.

Reviewer 2

1.Comment: After the revision, it became evident that the roots, leaves and stems are attributed to “growth regions”, while flowers and ripe fruits – to the “mature tissues”.

In this regard, I see the major issue: the conclusion that “the expression level of <MedWRKY> genes was much higher in the growth regions (root, stem, leaf) compared to the mature tissues (mature flower, ripe fruit), indicating that MedWRKYs may be crucial in controlling plant growth” does not correspond to the description of the MedWRKY gene expression presented in the section 2.6 (“MedWRKY14MedWRKY35, MedWRKY65, and MedWRKY114 were most highly expressed in ripe fruit and low in other tissues. MedWRKY32 was moderately expressed in stem, leave, and ripe fruit, and low in mature flower. MedWRKY59 was highly expressed in mature flower and ripe fruit, and low expression in other tissues. MedWRKY91 was highly expressed in root, leave and mature flower, and low expression was seen in stem and ripe fruit.”) I am not sure if it can be inferred from this text that WRKY expression is much higher in root, stem and leaves compared to flowers and fruits. Figures 8 and 9 also do not reveal this tendency.

1.Reply: In the section 2.6 (Line 192), we also analyzed the expression levels of genes in root “MedWRKY41, MedWRKY53, MedWRKY60, MedWRK61, MedWRK119, and MedWRKY125 were highly expressed in root and low in other tissues.” And according to the transcriptome data, we calculated the sum of the FPKM values of root, stem and leaf and the sum of the FPKM values of mature flower and ripe fruit, and the results showed that the expression level of the growth region was higher than that of the mature tissue. But the expression of this sentence is indeed a bit inaccurate, so we have deleted this sentence.

2. Comment:   Line 321: “most MedWRKY genes were mainly expressed in root”. This conclusion is also not supported by the data.

2. Reply: Thank you for your reminder. I'm sorry that my formulation of this sentence is inappropriate. We've changed this sentence to “The result of RT-qPCR and transcriptome data showed that some MedWRKY genes were expressed at the highest levels in the root, such as MedWRKY41, MedWRKY60, MedWRK61 and so on. This showed that these genes play a more important role in the root growth.” (Line 284)

3. Comment:   To my mind, a more relevant conclusion is presented in line 414: “Based on the transcriptome data and RT-qPCR data, we hypothesize that MedWRKY transcription factors play important roles in different tissues and different developmental periods of dodecandrum”.

However, here “transcription factors” should be replaced by “genes” since the authors analyzed the levels of MedWRKY transcripts, not proteins. The mRNA and protein expression patterns may significantly differ due to post-transcriptional regulation.

3.Reply: Thanks for your careful check. We've changed this sentence to “Based on the transcriptome data and RT-qPCR data, we hypothesize that MedWRKY genes play important roles in different tissues and different developmental periods of M.dodecandrum.”(Line 375)

4.Comment:Line 226: “The WRKY genes, which were not expressed at significant levels in other tissues, were likely to play a more general role in M.dodecandrum.” What “other tissues” did the authors mean?

4.Reply: Thank you for your reminder. According to the FPKM value, the expression level of the WRKY gene in stem is not significant. We've changed this sentence to “The WRKY genes, which were not expressed at significant levels in stem, were likely to play a more general role in M.dodecandrum.” (Line 203).

5.Comment:“The plant material used in this study came from the previous collection of the research group.” Please include a reference here (ref. 25?)

5.Reply: Thanks for your reminder. We've changed it to“The plant material used in this study came from the previous collection of the research group, and the genomic data and transcriptome data of M. dodecandrum used in this study came from Hao [25].”(Line 300)

6.Comment:In Figures 8 and 9: some red horizontal lines appear in the middle of the figures.

6.Reply: Thanks for your reminder. We don't know if it's because of the computer version, we don't see some red horizontal lines appear in the middle of the figures.

7.Comment: Figure 8, page 231, in the bottom: “big fruit” instead of “ripe fruit”. Gene names in the figure should be in Italics.

7.Reply: Thanks for your careful check. We have already made relevant corrections to the article. (Line 209)

Round 4

Reviewer 1 Report

Thank you authors for revising the manuscript.

However, I have the following comments for the authors. You need to revise it before it gets accepted. 

1.      In the abstract, line 21-23, the authors mentioned, “The results of promoter sequence analysis showed that there were many transcription factors binding sites and cis-acting elements related to plant growth and development, stress response, and secondary metabolite synthesis in the WRKY gene family.”

Actually, the cis-elements are the transcription factor binding sites. So, please revise the sentence. ….the promoter analysis identified a number of putative cis-elements most likely associated with transcription factors linked to the regulation of plant growth and development, stress response, and secondary metabolite synthesis in the WRKY gene family.

2.      Please revise the lines 25-27 “The study of the transcript group data revealed that 126 MedWRKY genes had variable levels of expression in various tissues and some MedWRKY genes were expressed at the highest levels in the root, suggesting that these genes play a more important role in root growth.

3.      In the Results, line 143-145, authors mentioned, “Analysis of the promoter sequences of 1500 bp upstream showed that many transcription factors binding sites related to plant growth and development, stress response, and secondary metabolite synthesis were present in the WRKY gene family (Table S2). “….please revise….showed the presence of many transcription factor binding sites associated with different transcription factors related to the regulation of …

Please check similar sentences throughout the manuscript and revise it.

4.      Revise Figure 6. It is not “0” and 3’. It is “TSS” and position “1”.

3’ lies downstream of TSS.

5.      In the Discussion, line 260-262, the authors mentioned, “In this study, we found that the promoter region of the MedWRKY genes contained multiple transcription factor binding  sites such as ZF-HD, bZIP, bHLH, etc. “. Please revise the sentence….transcription factor binding sites associated with transcription factors such as ZF-HD, bZIP, bHLH, etc…

6.      In the methods, line 337-340, the authors mentioned, “Then the transcription factor  sites to which promoter sequences may bind were analyzed at the Plant PAN (PlantPAN  4.0 (ncku.edu.tw)website, Arabidopsis thaliana was used as a reference species and select transcription factor binding sites with a score of 1.” Please revise the sentence. Actually, transcription factor binds to the cis-element or transcription factor binding site present in the promoter sequence.

7.      Please carefully check your language throughout the manuscript and revise.

Moderate editing of the language is required. The meaning of the sentence is wrong in many cases. 

Author Response

1、Comment:In the abstract, line 21-23, the authors mentioned, “The results of promoter sequence analysis showed that there were many transcription factors binding sites and cis-acting elements related to plant growth and development, stress response, and secondary metabolite synthesis in the WRKY gene family.”

Actually, the cis-elements are the transcription factor binding sites. So, please revise the sentence. ….the promoter analysis identified a number of putative cis-elements most likely associated with transcription factors linked to the regulation of plant growth and development, stress response, and secondary metabolite synthesis in the WRKY gene family.

1、Reply: Thank you for your reminder. The revised sentence is“The promoter sequence analysis identified a number of cis-acting elements related to plant growth and development, stress response, and secondary metabolite synthesis in the WRKY gene family.”(Line 21-23)

2、Comment:Please revise the lines 25-27 “The study of the transcript group data revealed that 126 MedWRKY genes had variable levels of expression in various tissues and some MedWRKY genes were expressed at the highest levels in the root, suggesting that these genes play a more important role in root growth.”

2、Reply:  Thank you for your reminder. The revised sentence is “The transcriptome data and RT-qPCR analysis suggested that MedWRKY genes had higher expression in root and ripe fruit of M. dodecandrum.” (Line 24-25)

3、Comment:In the Results, line 143-145, authors mentioned, “Analysis of the promoter sequences of 1500 bp upstream showed that many transcription factors binding sites related to plant growth and development, stress response, and secondary metabolite synthesis were present in the WRKY gene family (Table S2). “….please revise….showed the presence of many transcription factor binding sites associated with different transcription factors related to the regulation of …

Please check similar sentences throughout the manuscript and revise it.

3、Reply: Thank you for your reminder. We have made relevant modifications in Line 140-142. The revised sentence is “Analysis of the promoter sequences of 1500 bp upstream showed that the promoter cis-acting elements can bind different types of transcription factors, including bZIP, SBP, TCP and so on.”

4、Comment: Revise Figure 6. It is not “0” and 3’. It is “TSS” and position “1”. 3’ lies downstream of TSS

4、Reply: Thanks for your careful check.We have made relevant modifications in Line 163.

5、Comment: In the Discussion, line 260-262, the authors mentioned, “In this study, we found that the promoter region of the MedWRKY genes contained multiple transcription factor binding sites such as ZF-HD, bZIP, bHLH, etc. “. Please revise the sentence….transcription factor binding sites associated with transcription factors such as ZF-HD, bZIP, bHLH, etc…

5、Reply: Thank you for your reminder. The revised sentence is“In this study, we found that the promoter region of the MedWRKY genes contained multiple transcription factors binding sites associated with transcription factors ZF-HD, bZIP, bHLH, etc.”(Line 256-268)

6、Comment:In the methods, line 337-340, the authors mentioned, “Then the transcription factor sites to which promoter sequences may bind were analyzed at the Plant PAN (PlantPAN 4.0 (ncku.edu.tw) website, Arabidopsis thaliana was used as a reference species and select transcription factor binding sites with a score of 1.” Please revise the sentence. Actually, transcription factor binds to the cis-element or transcription factor binding site present in the promoter sequence.

6、Reply: Thank you for your suggestion. The revised sentence is“Then the promoter cis-acting elements were analyzed at the PlantPAN website (PlantPAN 4.0 (ncku.edu.tw)) and PlantCARE website (http://bioinformatics.psb.ugent.be/webtools/plantcare/html/). On the PlantPAN website, A. thaliana was used as a reference species and selected cis-acting element with a score of 1.” (Line 332-336)

7、Comment:Please carefully check your language throughout the manuscript and revise.

7、Reply: Thank you for your suggestion. We have invited a foreign friend who is fluent in English to help polish our article and made some revisions to the article. These changes will not influence the content and the framework of the paper. And here we did not list these changes.

Reviewer 2 Report

After the latest round of revision, I still encourage the authors to describe their gene expression data carefully.

1)    Lines 197-200: “MedWRKY41, MedWRKY53, MedWRKY60, MedWRK61, MedWRK119, and MedWRKY125 were highly expressed in root and low in other tissues. MedWRKY14, MedWRKY35, MedWRKY65, and MedWRKY114 were most highly expressed in ripe fruit and low in other tissues.”

According to this text from the “Results” section, six genes out of 126 were most highly expressed in the root, while four genes out of 126 were most highly expressed in the ripe fruit. The percentage of highly expressed genes is too low to allow any general conclusions to be drawn about any of these tissues. This applies to the conclusions in lines 26 and 284.

In my opinion, the authors should mention both tissues along with the highly expressed WRKY genes in their conclusions.

2)    Line 203: “The WRKY genes, which were not expressed at significant levels in stem, were likely to play a more general role in M.dodecandrum”. What do the authors mean by “significant levels”? Did you compare the expression levels statistically? I think the conclusion is that the expression levels of WRKY genes are generally lower in the stem compared to other tissues.

3)    Line 205: “Taken together, we speculate that the differences in gene expression may be related to the mechanism by which the MedWRKY genes regulate plant growth and development.” To my mind, this speculation came from the old version of the manuscript where the authors suggested that the WRKY genes were highly expressed in the “growing” tissues compared to the “mature” tissues. The “Results” should describe the data, speculations should be placed in the “Discussion” section.

4)    Line 195: “As shown in Figure 9, each MedWRKY gene was expressed in at least 1 tissue, except MedWRKY10 and MedWRKY42”. Since Figure 9 cannot visualize the low FPKM values (e.g. 0.005 for MedWRKY111, 0.05 for MedWRKY56, etc.), the authors should refer to Table S4 here.

Author Response

1.Comment:    Lines197-200: “MedWRKY41, MedWRKY53, MedWRKY60, MedWRK61MedWRK119, and MedWRKY125 were highly expressed in root and low in other tissues. MedWRKY14, MedWRKY35, MedWRKY65, and MedWRKY114 were most highly expressed in ripe fruit and low in other tissues.”

According to this text from the “Results” section, six genes out of 126 were most highly expressed in the root, while four genes out of 126 were most highly expressed in the ripe fruit. The percentage of highly expressed genes is too low to allow any general conclusions to be drawn about any of these tissues. This applies to the conclusions in lines 26 and 284.

In my opinion, the authors should mention both tissues along with the highly expressed WRKY genes in their conclusions.

1、Reply: Thank you for your suggestion. We have made relevant modifications in Line 200 and Line 280.

2、Comment:Line 203: “The WRKYgenes, which were not expressed at significant levels in stem, were likely to play a more general role in M.dodecandrum”. What do the authors mean by “significant levels”? Did you compare the expression levels statistically? I think the conclusion is that the expression levels of WRKY genes are generally lower in the stem compared to other tissues.

2、Reply: Thank you for your suggestion. “significant levels” refer to “the expression level of the WRKY gene is high”. I'm sorry that my formulation of this sentence is inappropriate. We think your expression is more accurate. The revised sentence is “Compared to other tissues, the MedWRKY genes had higher expression levels in the root and ripe fruit, and the expression levels of the MedWRKY genes in the stem were generally lower.” (Line 200-202)

3、Comment:Line 205: “Taken together, we speculate that the differences in gene expression may be related to the mechanism by which the MedWRKYgenes regulate plant growth and development.” To my mind, this speculation came from the old version of the manuscript where the authors suggested that the WRKY genes were highly expressed in the “growing” tissues compared to the “mature” tissues. The “Results” should describe the data, speculations should be placed in the “Discussion” section.

3、Reply:  Thank you for your suggestion. In “Results”, we removed this sentence and added the relevant description in “Discussion”.(Line 280)

4、Comment:Line 195: “As shown in Figure 9, each MedWRKYgene was expressed in at least 1 tissue, except MedWRKY10 and MedWRKY42”. Since Figure 9 cannot visualize the low FPKM values (e.g. 0.005 for MedWRKY111, 0.05 for MedWRKY56, etc.), the authors should refer to Table S4 here.

4、Reply: Thank you for your reminder. We have made relevant modifications in Line 190. The revised sentence is“As shown in Table S4, each MedWRKYgene was expressed in at least 1 tissue, except MedWRKY10 and MedWRKY42”.

Round 5

Reviewer 2 Report

During the last round of revision, the authors fully addressed all my comments.